# Presentation and validation of the Abbreviated Self Completion Teen-Addiction Severity Index (ASC T-ASI): A preference-based measure for use in health-economic evaluations

Vivian Reckers-Droog[1,2]*, Maartje Goorden[1¤], Yifrah Kaminer[3], Lieke van Domburgh[4,5], Werner Brouwer[1,2,6], Leona Hakkaart-van Roijen[1,2]

1 Erasmus Centre for Health Economics Rotterdam (EsCHER), Rotterdam, the Netherlands, 2 Erasmus University Rotterdam, Erasmus School of Health Policy & Management, Rotterdam, the Netherlands, 3 University of Connecticut School of Medicine, Alcohol Research Center and Injury Prevention Center, Farmington, United States of America, 4 VU University Medical Center, Department of Child and Adolescent Psychiatry, Amsterdam, the Netherlands, 5 Pluryn, Nijmegen, the Netherlands, 6 Erasmus University Rotterdam, Erasmus School of Economics, Rotterdam, the Netherlands

¤ Current address: Statistics Netherlands (CBS), The Hague, the Netherlands
* reckers@eshpm.eur.nl

## Abstract

Economic evaluations of new youth mental health interventions require preference-based outcome measures that capture the broad benefits these interventions can have for adolescents. The *Abbreviated Self Completion Teen-Addiction Severity Index (ASC T-ASI)* was developed to meet the need for such a broader measure. It assesses self reported problems in seven important domains of adolescents' lives, including school performance and family relationships, and is intended for use in economic evaluations of relevant interventions. The aim of the current study was to present the ASC T-ASI and examine its validity as well as its ability to distinguish between adolescents with and without problems associated with substance use and delinquency. The validation study was conducted in a sample of adolescents (n = 167) aged 12–18 years, who received in- or outpatient care in a youth mental health and (enclosed) care facility in the Netherlands. To examine its feasibility, test-retest reliability, and convergent validity, respondents completed the ASC T-ASI, as well as the EQ-5D-3L and SDQ at baseline and after a two-week interval using a counterbalanced method. The ASC T-ASI descriptive system comprises seven domains: substance use, school, work, family, social relationships, justice, and mental health, each expressing self reported problems on a five-point Likert scale (ranging from having 'no problem' to having a 'very large problem'). The majority of respondents (>70%) completed the ASC T-ASI within 10 minutes and appraised the questions as (very) easy and (very) comprehensible. Test-retest reliability was adequate ($K_w$ values 0.26–0.55). Correlations with the supplementary measures were moderate to high ($r_s$ 0.30–0.50), suggesting convergent validity. The ASC T-ASI is a promising and valid measure for assessing self reported problems in important domains in adolescents' lives, allowing benefits beyond health and health-related quality of

**Data Availability Statement:** All relevant data are within the manuscript and its Supporting Information files.

**Funding:** Funding for the validation study was obtained by LHR from the Netherlands Organisation for Health Research and Development (ZonMW; https://www.zonmw.nl/nl/), project number 157004007. The funders had no role in the study design, data collection and analysis, interpretation of the data, writing of the manuscript, and/or decision to submit the manuscript for publication. The views expressed in this article are those of the authors.

**Competing interests:** The authors have declared that no competing interests exist.

life to be included in economic evaluations of youth mental health interventions. Future studies of the ASC T-ASI should consider the comprehensiveness of its domains and sensitivity to change.

## Introduction

Economic evaluations of new health interventions are increasingly used by decision makers to inform reimbursement decisions in health systems around the world [1]. In the majority of health-economic guidelines [2–4], use of the Quality-Adjusted Life-Year (QALY) is recommended as it integrates gains in length and quality of life into a single generic outcome, allowing comparisons across disease areas and patient populations [5–7]. QALYs are typically estimated through the use of generic preference-based measures such as the EQ-5D, Short Form 6 Dimensions (SF-6D), and Health Utility Index Mark 3 (HUI-3) [8–10]. In recent years, the EQ-5D has become the preferred measure in various European countries, including the Netherlands and United Kingdom (UK) [2,3,11]. The EQ-5D measures health-related quality of life (HRQOL) in the domains: mobility, self-care, usual activities, pain/discomfort, and depression/anxiety [8].

Recently, several generic preference-based measures have been developed that enable economic evaluations of new health technologies for children and adolescents. For example, the EQ-5D-Y (for which a preference-based value set is not yet available) and the Child Health Utility instrument (CHU9D) [12,13]. Although these generic measures can be used to estimate QALY gains and facilitate the allocation of scarce healthcare resources across patient populations and competing health interventions [5], there is growing recognition of the relevance of treatment-related benefits beyond health and HRQOL as captured by common QALY measures. In the treatment of adolescents with mental health problems, including those with problems associated with substance use and delinquency, concerns have been raised regarding the sensitivity and comprehensiveness of generic preference-based measures, such as the EQ-5D, in capturing all relevant treatment benefits [7,13–16]. For example, the EQ-5D has been found to be unresponsive to health changes in some populations with more complex mental health problems [17,18]. In addition, youth mental health interventions do not necessarily aim to improve (only) health or HRQOL, but rather focus on benefits beyond that scope. Such benefits may include decreasing adolescents' use of substances, reducing their problems with the juvenile justice system, and improving their educational achievements (note that the latter is incorporated in the CHU9D domain schoolwork) [13,19–22]. Moreover, currently available preference-based measures typically focus on treatment benefits that may affect the individual patient, such as their mobility and ability to do their schoolwork [8,12,13]. However, youth mental health interventions generally yield benefits that may not only affect the individual patient but may also have "spill-over effects" on their friends and family [21–23]. Indeed, youth mental health interventions may specifically focus on improving the interactions between adolescents and their parents, siblings, other family members, peers, and neighbours [21,22]. As generic preference-based measures do not capture these broader benefits, the benefits of youth mental health interventions may be underestimated when using conventional outcome measures in economic evaluations [17]. Consequently, the incremental cost-effectiveness ratios (ICERs) may not reflect the value for money offered by these interventions, which may lead to undesirable outcomes of associated reimbursement decisions [24].

This study reports on the development and validation of the *Abbreviated Self Completion Teen-Addiction Severity Index (ASC T-ASI)* that was developed in order to better meet the need for a preference-based measure for assessing the (cost-) effectiveness of youth mental health interventions. The aim of the current study was to present the newly developed ASC T-ASI, describe its development process, and examine its validity in a broad population of adolescent mental health patients, including those with problems associated with substance use and delinquency.

## Materials and methods

### Development of the ASC T-ASI

The ASC T-ASI was developed and validated in three consecutive steps. First, the T-ASI was identified as a suitable measure for capturing the broad benefits of youth mental health interventions [14]. The self reported items were selected from the domains of the T-ASI to generate the ASC T-ASI descriptive system. Second, a validation study was conducted to examine the feasibility, test-retest reliability, and convergent validity of the ASC T-ASI. Third, a discrete choice experiment was performed to derive preference weights (tariffs) from the Dutch general population for the different states described by the ASC T-ASI. That study is described elsewhere [25], but the derived tariffs are used in the current study. In this article, we present the ASC T-ASI, describe its development process, and report on the design and results of the validation study.

### Item generation

Schawo et al. [14] conducted a systematic literature review and selected the (non-preference based) Teen-Addiction Severity Index (T-ASI) [26] as a comprehensive measure for capturing the broad benefits of youth mental health interventions. The T-ASI covers the domains substance use, school, work, family, social relationships, justice, and mental health. The T-ASI is normally completed jointly by a healthcare professional and the patient, with some questions directed at the patient. The ASC T-ASI descriptive system was developed by selecting those patient–oriented questions that cover self reported problems in each of the T-ASI domains. Answering categories consist of a five level scale, ranging from having 'no problem' to having a 'very large problem' in each of seven domains. The ASC T-ASI was translated into the Dutch language and converted to B1 reading level by a professional translation agency in order to make it easy to read and understand for adolescents, also given the intended target group.

### Validation study

**Sample and design.** The data were collected between October 2015 and February 2016, in a sample of 171 adolescents who received in- or outpatient care in a youth mental health and (enclosed) care facility in the Netherlands. Respondents were 12–18 years old and literate in Dutch. According to the Medical Research Involving Human Subjects Act [27], no ethical approval was required for this study. To be certain of this, we submitted our study protocol to the Research Ethics Review Committee of the Erasmus School of Health Policy & Management and received written confirmation (reference IRB 2020–04 WMO). Prior to participation, we informed respondents and their parents or legal guardians about the study objective and invited them to participate in the study. We explained to respondents that their participation was voluntary and their anonymity would be ensured. We also explained that they could withdraw from the study at any moment, in which case their data would be discarded. Respondents could enter the study only after giving written consent for their participation and the use of

their data for research purposes prior to taking part in the study. When adolescents were aged between 12 and 16 years, additional written consent was obtained from their parents or legal guardians. The questionnaires were administered to respondents by sociotherapists or teachers known to them. Before respondents completed the questionnaire, the sociotherapists and teachers again explained to respondents that they could withdraw from the study at any moment and asked them to confirm verbally that they were (still) willing to participate.

At baseline ($T_0$), respondents completed questions relating to their socio-demographic characteristics, followed by a test-questionnaire that included the ASC T-ASI, EQ-5D 3 Levels (EQ-5D-3L), and Strengths and Difficulties Questionnaire (SDQ). The latter two measures are further introduced below. To reduce the risk of ordering bias, we used a counterbalancing method and randomly assigned respondents to complete the test-questionnaire in one of six orders (Q1 to Q6). To examine the test-retest reliability of the ASC T-ASI, a subsample of 32 respondents completed a retest-questionnaire after a two-week interval ($T_1$). All but one of these respondents completed the retest-questionnaire in the same order as at $T_0$.

**Outcome measures.** The *EQ-5D-3L* [8] is a validated, standardised generic preference-based measure that assesses self reported HRQOL 'today'. The EQ-5D-3L descriptive system comprises five dimensions: mobility, self-care, usual activities, pain/discomfort, and anxiety/depression. In each of the dimensions, self reported problems are expressed on a three level scale, resulting in 243 ($3^5$) unique five-digit EQ-5D-3L codes. Each code can be transformed into a summary utility index, normalized so that 0 represents the utility of the state 'dead' and 1 that of 'full health'. Health states considered as 'worse than dead' are valued below 0. All respondents completed the adult version of the EQ-5D-3L, which is suitable for use from the age of 12 onwards [8].

The *Strengths and Difficulties Questionnaire (SDQ)* [28] is a validated, 25-item behavioural screening questionnaire, measuring psychological well-being in children and adolescents 'during the last six months'. The SDQ comprises five dimensions: emotional symptoms, conduct problems, hyperactivity/inattention, peer relationship problems, and prosocial behaviour, for which a total 'difficulties score' between 0 and 40 can be calculated. The SDQ difficulties score is calculated by summing scores from four dimensions, excluding the dimension prosocial behaviour. All respondents completed the extended self-report version for 11–17 year olds.

## Statistical analyses

Descriptive statistics of the sample, including the mean (SD) ASC T-ASI tariff, were calculated at baseline. Tariffs were based on relative preferences that were elicited from a representative sample of the Dutch general adult population [29]. The ASC T-ASI tariff is calculated by converting the scores on the seven domains into a single summary index score, using an additive function. The ASC T-ASI tariff ranges between 0 and 1. A tariff of 0 refers to the worst state and a tariff of 1 to the best possible state as defined by the ASC T-ASI [24]. Differences in ASC T-ASI and EQ-5D-3L tariff, and in SDQ difficulties score between patients who did and did not use substances (i.e. alcohol, drugs, and/or medicines) and/or conducted delinquent acts were examined using independent t-tests. A Bonferroni correction was applied to adjust for the increased risk of a Type 1 error, caused by multiple comparisons.

Feasibility of the ASC T-ASI was investigated at $T_0$ by assessing the time to complete the ASC T-ASI, the level of difficulty and comprehensiveness of the questions, and the percentage of respondents with missing values. A missing value was defined as a respondent checking none or more than the one answer option that was required per domain. Beforehand, we decided that completing the ASC T-ASI should not take more than 10 minutes and that at least 70% of respondents should appraise the questions as being (very) comprehensible and

(very) easy to answer. Test-retest reliability of the ASC T-ASI was investigated by calculating the percentage of complete agreement between $T_0$ and $T_1$ scores and linear weighted Kappa ($K_w$) coefficients for $T_0$ and $T_1$ scores (presented with 95% CIs). In accordance with Landis & Koch's standards for strength of agreement [30], $K_w < 0$ was interpreted as poor agreement, 0–0.20 as slight agreement, 0.21–0.40 as fair agreement, 0.41–0.60 as moderate agreement, 0.61–0.80 as substantial agreement, and 0.81–1.00 as indicating almost perfect agreement. Convergent validity of the ASC T-ASI was investigated at $T_0$ by (i) comparing scores of the ASC T-ASI domain mental health to the EQ-5D-3L dimension anxiety/depression and the SDQ difficulties score, (ii) comparing scores of the ASC T-ASI domain social relationships to the SDQ dimension peer relationship problems, and (iii) comparing the ASC T-ASI tariff to the EQ-5D-3L tariff and SDQ difficulties score, using Spearman's rank-correlation coefficients ($r_s$; Bonferroni corrected). In accordance with Cohen's [31] classification system, we interpreted correlations below 0.30 as low, between 0.30 and 0.50 as moderate, and above 0.50 as high. Moderate $r_s$ values were considered to be a sign of convergent validity of the ASC T-ASI since the ASC T-ASI measures a related, yet broader construct than the EQ-5D-3L and SDQ.

We conducted the analyses using Stata 16.0 (Stata Corp LP, College station, Texas).

## Results

### Abbreviated self completion teen-addiction severity index

Following the T-ASI, the descriptive system of the ASC T-ASI comprises seven domains: substance use, school, work, family, social relationships, justice, and mental health. In each of the ASC T-ASI domains, self reported problems are expressed on a five level scale ranging from having 'no problem' to having a 'very large problem'. The descriptive system is able to distinguish 78,125 ($5^7$) unique ASC T-ASI states, represented by seven-digit ASC T-ASI codes [29]. Table 1 presents the ASC T-ASI descriptive system (English version, B1 level), while the Dutch version is included as S1 Text.

### Validation study

Table 2 presents the characteristics of the sample (n = 167) that remained after excluding respondents who did not meet the age requirements (n = 3) and checked the first answer option of all questions included in the test-questionnaire (n = 1). Mean (SD) age of respondents was 15.2 (1.7) years. The majority of respondents (n = 142; 85.0%) was Dutch and received inpatient mental health care (n = 99; 59.3%) for a range of mental health problems. Of the respondents, 55 (32.9%) used alcohol and 32 (19.2%) used drugs, of whom 26 (81.3%) used soft drugs (defined in terms of drugs that may cause psychological addiction, e.g. hash and marijuana [32]) and 6 (18.7%) used a combination of soft and hard drugs (the latter defined in terms of drugs that may cause psychological *and* physical addiction, e.g. cocaine and amphetamine [32]). Of the 91 (54.5%) respondents who used medicines, 49 (53.8%) used psychopharmaceuticals, e.g. for treatment of anxiety, depression, psychosis, Attention Deficit Hyperactivity Disorder (ADHD), Pervasive Developmental Disorder-Not Otherwise Specified (PDD-NOS), Multiple Complex Developmental Disorder (MCDD), and Oppositional Defiant Disorder (ODD) (not in table). A total of 68 (40.7%) respondents had been in contact with the judicial authorities as a result of delinquent conduct.

**ASC T-ASI codes.** We distinguished 91 unique ASC T-ASI states in our sample. The mean (SD) ASC T-ASI tariff was 0.89 (0.10) and ranged from 0.55 to 1.00. Fig 1 presents the frequency (%) of the problems reported by respondents in each of the ASC T-ASI domains. Of the respondents, 121 (72.5%) reported having at least a 'slight problem' in one or more domains. Specifically, 87 (52.1%) reported having a 'slight problem' in the domain school,

**Table 1. Abbreviated self completion teen-addiction severity index (English version, B1 level).**

| Please check the answer that currently fits you best: | |
|---|---|
| **1**. <u>Substance use</u> | |
| I have **no problem** with the use of alcohol, drugs or medicine | ❏ |
| I have a **slight problem** with the use of alcohol, drugs or medicine | ❏ |
| I have a **fairly large problem** with the use of alcohol, drugs or medicine | ❏ |
| I have a **large problem** with the use of alcohol, drugs or medicine | ❏ |
| I have a **very large problem** with the use of alcohol, drugs or medicine | ❏ |
| **2**. <u>School</u> | |
| I have **no problem** with school | ❏ |
| I have a **slight problem** with school | ❏ |
| I have a **fairly large problem** with school | ❏ |
| I have a **large problem** with school | ❏ |
| I have a **very large problem** with school | ❏ |
| **3**. <u>Work</u> | |
| I have **no problem** with work | ❏ |
| I have a **slight problem** with work | ❏ |
| I have a **fairly large problem** with work | ❏ |
| I have a **large problem** with work | ❏ |
| I have a **very large problem** with work | ❏ |
| **4**. <u>Family</u> | |
| I have **no problem** with family | ❏ |
| I have a **slight problem** with family | ❏ |
| I have a **fairly large problem** with family | ❏ |
| I have a **large problem** with family | ❏ |
| I have a **very large problem** with family | ❏ |
| **5**. <u>Social relationships</u> | |
| I have **no problem** with friends, acquaintances and others in my environment | ❏ |
| I have a **slight problem** with friends, acquaintances and others in my environment | ❏ |
| I have a **fairly large problem** with friends, acquaintances and others in my environment | ❏ |
| I have a **large problem** with friends, acquaintances and others in my environment | ❏ |
| I have a **very large problem** with friends, acquaintances and others in my environment | ❏ |
| **6**. <u>Justice</u> | |
| I have **no problem** with the judicial authorities | ❏ |
| I have a **slight problem** with the judicial authorities | ❏ |
| I have a **fairly large problem** with the judicial authorities | ❏ |
| I have a **large problem** with the judicial authorities | ❏ |
| I have a **very large problem** with the judicial authorities | ❏ |
| **7**. <u>Mental health</u> | |
| I have **no problem** with my mental health | ❏ |
| I have a **slight problem** with my mental health | ❏ |
| I have a **fairly large problem** with my mental health | ❏ |
| I have a **large problem** with my mental health | ❏ |
| I have **a very large problem** with my mental health | ❏ |

followed by 74 (44.9%) in family, 50 (29.9%) in mental health, 44 (26.3%) in social relationships, 40 (24.0%) in justice, 29 (17.4%) in substance use, and 19 (11.4%) in the work domain. Of the respondents, 24 (14.4%) reported having 'no problem' in any of the domains, compared to 55 (32.9%) reporting having 'no problem' in any of the EQ-5D-3L dimensions, and 0 (0.0%) in any of the SDQ dimensions.

**Table 2. Sample characteristics ($n$ = 167)[a].**

|  | Mean (SD) or % | Min | Max |
|---|---|---|---|
| Age (Years) | 15.2 (1.7) | 12 | 18 |
| Gender (Female) | 50.9 | | |
| Ethnicity (Dutch) | 85.0 | | |
| Inpatient | 59.3 | | |
| Attending school | 91.0 | | |
| Playing sports | 53.3 | | |
| Working | 21.6 | | |
| Physical illness/ handicap | 20.4 | | |
| Smoking | 43.7 | | |
| Substance use | | | |
| Alcohol | 32.9 | | |
| Drugs | 19.2 | | |
| Medicine use | 54.5 | | |
| Delinquency | 40.7 | | |
| ASC T-ASI tariff | 0.89 (0.10) | 0.55 | 1.00 |
| EQ-5D-3L tariff | 0.82 (0.20) | -0.33 | 1.00 |
| SDQ difficulties score | 13.82 (5.69) | 2.00 | 29.00 |

ASC T-ASI, Abbreviated Self Completion Teen-Addiction Severity Index; EQ-5D-3L, EQ-5D 3 Levels; SDQ, Strengths and Difficulties Questionnaire.

[a] Number of respondents (%) per questionnaire version: 25 (15.0%) for Q1, 32 (19.2%) for Q2, 28 (16.8%) for Q3, 27 (16.2%) for Q4, 34 (20.4%) for Q5, and 21 (12.6%) for Q6.

Table 3 presents the differences in mean (SD) ASC T-ASI and EQ-5D-3L tariffs, and SDQ score between respondents who did (not) use substances and/or conducted delinquent acts. The ASC T-ASI tariff was lower for respondents who used alcohol and drugs, and were delinquent than than for those who did not use these substances and were not delinquent. Independent t-tests (Bonferroni corrected, two-tailed, $\alpha/4$) revealed that these differerences were statistically significant at the 0.01 level for alcohol and drug use, and at the 0.001 level for delinquency. For comparison, the EQ-5D-3L tariff and SDQ difficulties score did not statistically significantly differ between these subgroups.

**Feasibility.** Of the respondents, 17 (10.2%) had a missing value in one or more ASC T-ASI domains and 5 (3.0%) checked more than the one required answer option per domain. The missing values per domain ranged from 0.0% to 7.8%, with social relationships having the least and work having the most most missing values.

The majority of respondents (n = 128; 78.5%) completed the ASC T-ASI within 10 minutes, with a mode of 0–5 minutes (n = 66; 40.5%), and appraised the questions as being (very) easy (n = 124; 76.1%) and (very) comprehensible (n = 121; 75.6%). Few respondents appraised the questions as being (very) difficult (n = 8; 4.8%) and (very) incomprehensible (n = 9; 5.4%).

**Test-retest reliability.** A subsample of respondents (n = 32) completed the retest-questionnaire on average 20.3 days (SD = 5.9, range 13–31 days) after assessing the test-questionnaire.

Table 4 presents test-retest reliability statistics for each of the ASC T-ASI domains. The percentages of complete agreement ranged from 54.8% to 81.3%. $K_w$ values ranged from 0.26 to 0.55, indicating fair to moderate agreement between $T_0$ and $T_1$ scores on the ASC T-ASI domains.

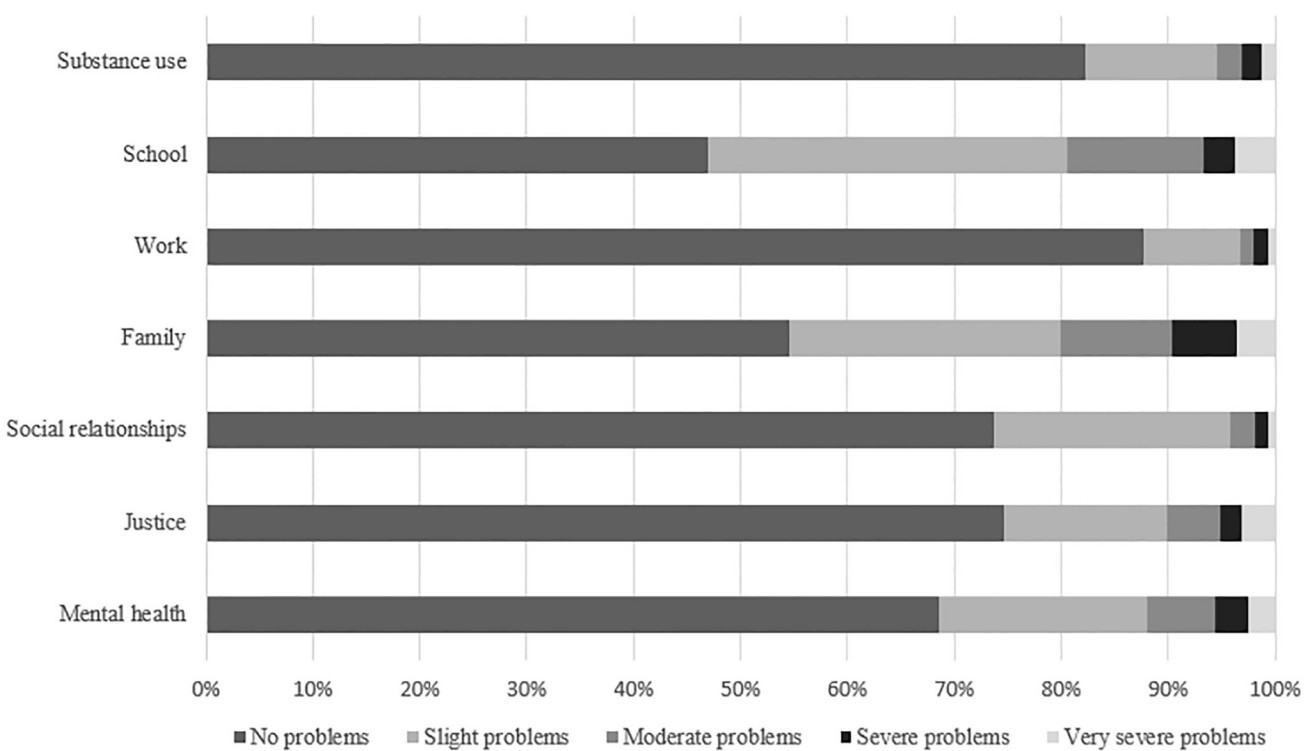

**Fig 1. Frequency of problems reported by respondents (in % of total) in the ASC T-ASI domains.**

**Convergent validity.** Positive correlations of moderate strength were found between the ASC T-ASI domain mental health and the EQ-5D-3L dimension anxiety/depression ($r_s = 0.304$) and SDQ difficulties score ($r_s = 0.466$). Positive correlations of moderate strength were also found between the ASC T-ASI domain social relationships and the SDQ dimension peer relationship problems ($r_s = 0.408$) and between the ASC T-ASI and EQ-5D-3L tariffs ($r_s = 0.350$). A negative correlation of moderate strength was found between the ASC T-ASI tariff

**Table 3. Mean (SD) ASC T-ASI and EQ-5D-3L tariffs, and SDQ difficulties score of adolescents who do (not) use substances and/or are (not) delinquent[a].**

| | ASC T-ASI[b] | | | | | EQ-5D-3L | | | | | SDQ | | | | |
|---|---|---|---|---|---|---|---|---|---|---|---|---|---|---|---|
| | SUD | | no SUD | | | SUD | | no SUD | | | SUD | | no SUD | | |
| | *n* | Mean (SD) | *n* | Mean (SD) | p-value | *n* | Mean (SD) | *n* | Mean (SD) | p-value | *n* | Mean (SD) | *n* | Mean (SD) | p-value |
| Substance use | | | | | | | | | | | | | | | |
| Alcohol | 51 | 0.86 (0.11) | 90 | 0.91 (0.09) | 0.001** | 52 | 0.82 (0.23) | 105 | 0.82 (0.18) | 0.878 | 46 | 14.63 (5.65) | 89 | 13.30 (5.52) | 0.197 |
| Drugs | 28 | 0.83 (0.12) | 115 | 0.91 (0.09) | 0.001** | 31 | 0.83 (0.18) | 128 | 0.82 (0.20) | 0.696 | 24 | 14.00 (5.63) | 112 | 13.79 (5.72) | 0.868 |
| Medicine use | 76 | 0.89 (0.11) | 65 | 0.89 (0.10) | 0.869 | 86 | 0.81 (0.18) | 71 | 0.84 (0.22) | 0.325 | 71 | 14.18 (5.24) | 63 | 13.37 (6.21) | 0.410 |
| Delinquency | 61 | 0.85 (0.11) | 73 | 0.93 (0.07) | 0.000*** | 67 | 0.82 (0.23) | 82 | 0.82 (0.18) | 0.907 | 54 | 13.56 (5.76) | 73 | 13.77 (5.63) | 0.836 |

ASC T-ASI, Abbreviated Self Completion Teen-Addiction Severity Index; SDQ, Strengths and Difficulties Questionnaire; SUD, substance use and/or delinquency; no SUD, no substance use and/or delinquency.

[a] Note that the numbers of respondents do not correspond between the outcome measures due to differences in missing values.

[b] Note that the numbers of respondents do not correspond with the presented sample characteristics in Table 2 due to missing values in one or more of the ASC T-ASI domains.

** p-value ≤ 0.01,

*** p-value ≤ 0.001 (after Bonferroni correction, α/4).

**Table 4. Test-retest reliability of ASC T-ASI domains in percentage of complete agreement and $K_w$ coefficients[a].**

| Domain | n | Complete agreement (%) | $K_w$ value | 95% CI $K_w$ value | |
|---|---|---|---|---|---|
| | | | | Lower bound | Upper bound |
| Substance use | 32 | 81.3 | 0.30 | -0.04 | 0.68 |
| School | 31 | 67.7 | 0.55 | 0.21 | 0.79 |
| Work | 27 | 77.8 | 0.35 | -0.07 | 0.78 |
| Family | 32 | 62.5 | 0.47 | 0.19 | 0.70 |
| Social relationships | 31 | 64.5 | 0.26 | -0.05 | 0.55 |
| Justice | 30 | 70.0 | 0.49 | -0.10 | 0.80 |
| Mental health | 31 | 54.8 | 0.32 | 0.01 | 0.61 |

ASC T-ASI, Abbreviated Self Completion Teen-Addiction Severity Index; $K_w$, linear weighted kappa.

[a] Measured after a two-week interval.

and the SDQ difficulties score ($r_s$ = -0.503). All correlations were statistically significant at the 0.01 level (two-tailed, Bonferroni corrected, $\alpha/5$).

## Discussion and conclusions

This study presented the ASC T-ASI as a self reported preference-based measure that captures benefits beyond health and HRQOL that can be used in economic evaluations of mental health interventions, including those aimed at adolescents with problems associated with substance use and delinquency. The results of the validation study are promising and indicate that the ASC T-ASI is a relatively quick, easy, and comprehensible measure to complete for adolescents. The ASC T-ASI showed an adequate test-retest reliability and a convergent validity of moderate strength. In addition, the results indicate that the ASC T-ASI is sensitive to differences between adolescents who do and do not use substances and/or conduct delinquent acts, where the EQ-5D-3L and SDQ appear not not to be. Furthermore, respondents most frequently reported problems in the ASC T-ASI domains school and family. These domains are not (directly) covered by the EQ-5D-3L (nor by the SDQ), and, consequently, benefits in these domains may currently be neglected in economic evaluations of youth mental health interventions. This emphasises the importance of new comprehensive measures, like the ASC T-ASI.

A strength of the ASC T-ASI development process concerns the systematic selection of the T-ASI as a basis for developing the ASC T-ASI. Although the T-ASI was selected based on its ability to capture the broad benefits of youth mental health interventions, this does not necessarily imply that the T-ASI captures *all* relevant treatment benefits. Therefore, future research of the ASC T-ASI should focus on establishing the comprehrensiveness of its domains, also in different contexts. A strength of the validation study concerns the broad range of mental health problems of the adolescents included in our sample. This width increases the generalisabily of our results to a broad population of adolescent mental health patients in the Netherlands [33,34]. Some limitations also need to be discussed. A first limitation concerns our use of the EQ-5D-3L, rather than the EQ-5D-5L, for investigating the convergent validity of the ASC T-ASI. At the time of study design and data collection, the Dutch EQ-5D-5L preference-based value set was not published [35]. Moreover, it is uncertain whether using the EQ-5D-5L rather than the EQ-5D-3L would have substantially affected the results and conclusions drawn. The EQ-5D-5L is expected to be more sensitive than the EQ-5D-3L in the health domain, but this does not imply better differentiation between 'having' or 'not having' problems in broader domains [36]. A second limitation concerns the only fair test-retest reliability of four ASC T-ASI domains, i.e. substance use, work, social relationships, and mental health. Differences in

the domain scores may reflect therapeutic changes during the test-retest interval. However, this was not anticipated beforehand, and hence the retest-questionnaire did not include any additional questions by means of which we could have (partially) explained discordances between $T_0$ and $T_1$ scores in the ASC T-ASI domains. For example, a question relating to respondent's relationship with peers could have enabled us to interpret the change in the domain social relationships between $T_0$ and $T_1$. Future research should focus on clarifying whether differences between these ASC T-ASI domain scores result from their limited reliability or from their sensitivity to change. In line with the previous limitation, we consider it a third, and final, limitation that no longterm data were available for examining the ASC T-ASI's sensitivity to change. Indeed, when assessing the (cost-) effectiveness of youth mental health interventions with the use of the ASC T-ASI, it is important that this measure is sensitive to changes in mental health problems over time. Future research of the ASC T-ASI should therefore also focus on examining this psychometric property. In addition, future research should be focused on confirming the current promising results in larger and other samples of adolescents who may benefit from youth mental health interventions, in different contexts, and potentially in other countries.

The results of this first validation of the ASC T-ASI suggest that is a promising and valid preference-based measure for assessing self reported problems in multiple important domains in adolescents' lives. Ultimately, measures like this could allow benefits beyond health and HRQOL to be included in economic evaluations of youth mental health patients. Next to further validation, this also requires a clear view of how inclusion of such an measure in economic evaluations should take place (e.g. instead of or next to current HRQOL measures and certain cost types, etc.) and how results should be interpreted and used, for example, in reimbursement decisions. This validation study provides a good base for further research on both the comprehensiveness and the validity of the ASC T-ASI for application in (cost-) effectiveness research concerning the treatment of adolescents in youth mental healthcare.

## Supporting information

**S1 Text. Abbreviated self completion teen-addiction severity index (Dutch version, B1 level).**
(DOCX)

**S1 Data. Dataset of the study variables.**
(CSV)

**S2 Data.**
(XLSX)

## Author Contributions

**Conceptualization:** Vivian Reckers-Droog, Maartje Goorden, Yifrah Kaminer, Leona Hakkaart-van Roijen.

**Data curation:** Vivian Reckers-Droog.

**Formal analysis:** Vivian Reckers-Droog.

**Funding acquisition:** Leona Hakkaart-van Roijen.

**Investigation:** Vivian Reckers-Droog, Maartje Goorden, Lieke van Domburgh, Leona Hakkaart-van Roijen.

**Methodology:** Vivian Reckers-Droog, Maartje Goorden, Leona Hakkaart-van Roijen.

**Project administration:** Vivian Reckers-Droog, Leona Hakkaart-van Roijen.

**Supervision:** Werner Brouwer, Leona Hakkaart-van Roijen.

**Validation:** Vivian Reckers-Droog, Werner Brouwer, Leona Hakkaart-van Roijen.

**Visualization:** Vivian Reckers-Droog.

**Writing – original draft:** Vivian Reckers-Droog.

**Writing – review & editing:** Vivian Reckers-Droog, Maartje Goorden, Yifrah Kaminer, Lieke van Domburgh, Werner Brouwer, Leona Hakkaart-van Roijen.

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
