## [Decision Letter · Decision Letter 0]

28 Jul 2020

PONE-D-20-03462

Presentation and Validation of the Abbreviated Self Completion Teen-Addiction Severity Index (ASC T-ASI): A Preference-Based Measure for Use in Health-Economic Evaluations

PLOS ONE

Dear Dr. Reckers-Droog,

Thank you for submitting your manuscript to PLOS ONE. After careful consideration, we feel that it has merit but does not fully meet PLOS ONE’s publication criteria as it currently stands. Therefore, we invite you to submit a revised version of the manuscript that addresses the points raised during the review process.

We look forward to receiving your revised manuscript.

Kind regards,

Matthew J. Gullo

Academic Editor

PLOS ONE

Journal Requirements:

Additional Editor Comments:

1.) Please report group comparisons for EQ-5D-3L and SDQ in Table 2.

2.) Test-retest reliablity is only slight/fair for most scales and this should come through more clearly in the Discussion. While it may be that these scores reflect substantial therapeutic change occurring during the test-retest interval, this was not anticipated a priori in the study design. It may be that low reliability is a limitation of some of the included scales. Future research could seek to clarify the matter.

Reviewers' comments:

Reviewer's Responses to Questions

**Comments to the Author**

1. Is the manuscript technically sound, and do the data support the conclusions?

Reviewer #1: Yes

2. Has the statistical analysis been performed appropriately and rigorously? 

Reviewer #1: Yes

3. Have the authors made all data underlying the findings in their manuscript fully available?

Reviewer #1: Yes

4. Is the manuscript presented in an intelligible fashion and written in standard English?

Reviewer #1: Yes

5. Review Comments to the Author

Reviewer #1: The current study addresses an important area of increasing prominence, that of accurate economic evaluation of mental health and substance use programs. The authors outline and validate a new measure intending to capture broad ranging treatment benefits of mental health interventions, that may be missed by existing scales. The authors clearly outline the scale development process. However, I am concerned about the lack of ethical approval to conduct this study. I note the authors said they did consult an IRB, did they provide approval for the research to be undertaken? If no I think further justification is needed.

More specific comments are provided below to improve the manuscript:

- The authors should consider mentioning the Child Health Utility 9-D in the introduction as a previous economic scale used frequently in mental health program economic evaluation and explain what their scale adds above this existing scales, which covers most of the domains in the current scale (except addressing substance use directly).

On page 8 the authors should define what they mean by the terms 'soft' and 'hard drugs'

The results section would be improved if the authors reported on the direction of the results, as well as significant differences.

I note the authors conducted a bonferonni correction to control error rate however given the number of comparisons why was an ANOVA not used?

- I was confused by the statement that a limitation of the study were the lack of questions in the retest questionnaire, i had thought the same questionnaire was delivered at retest?

6. PLOS authors have the option to publish the peer review history of their article (what does this mean?). If published, this will include your full peer review and any attached files.

Reviewer #1: No

---

## [Author Response · Author response to Decision Letter 0]

17 Aug 2020

Manuscript: PONE-D-20-03462

Manuscript title: Presentation and validation of the Abbreviated Self Completion Teen-Addiction Severity Index (ASC T-ASI): A preference-based measure for use in health-economic evaluations

RESPONSE TO REVIEWERS

We thank the editor and reviewer for their positive remarks and the opportunity to submit a revised version of our manuscript. We have carefully revised the manuscript in view of the constructive and helpful revision suggestions as outlined in detail below. Please note that we respond to the journal requirements, editor, and reviewer #1 in the order of the received comments. In our response, we refer to the pages and line numbers of the Revised Manuscript with Track Changes.

JOURNAL REQUIREMENTS

Response: As requested, we have revised our manuscript and file names to meet PLOS ONE’s style requirements.

RESPONSE TO THE EDITOR

1. Please report group comparisons for EQ-5D-3L and SDQ in Table 2.

Response: In line with the editor’s comment, we now also report comparisons between respondents who do (not) use substances and are (not) delinquent for the EQ-5D-3L and SDQ in Table 2.

2. Test-retest reliability is only slight/fair for most scales, and this should come through more clearly in the Discussion. While it may be that these scores reflect substantial therapeutic change occurring during the test-retest interval, this was not anticipated a priori in the study design. It may be that low reliability is a limitation of some of the included scales. Future research could seek to clarify the matter.

Response: In line with the editor’s comment, we now discuss the reliability of four ASC T-ASI domains in the limitations section of the Discussion (page 13, lines 278-283). 

RESPONSE TO REVIEWER #1

1. The current study addresses an important area of increasing prominence, that of accurate economic evaluation of mental health and substance use programs. The authors outline and validate a new measure intending to capture broad ranging treatment benefits of mental health interventions, that may be missed by existing scales. The authors clearly outline the scale development process. However, I am concerned about the lack of ethical approval to conduct this study. I note the authors said they did consult an IRB, did they provide approval for the research to be undertaken? If no I think further justification is needed.

Response: We thank the reviewer for the useful comments and the positive evaluation of our work. In line with the reviewer’s comment, we included a reference to the Medical Research Involving Human Subjects Act in the Netherlands, elaborated on our consultation with the Research Ethics Review Committee of the Erasmus School of Health Policy & Management and on the procedure that we followed to obtain written as well as verbal consent of the respondents (see pages 4 and 5, lines 99-112).

2. The authors should consider mentioning the Child Health Utility 9-D in the introduction as a previous economic scale used frequently in mental health program economic evaluation and explain what their scale adds above this existing scales, which covers most of the domains in the current scale (except addressing substance use directly).

Response: Following the reviewer’s comment, we now also mention the CHU9D as well as the EQ-5D-Y in the Introduction section and explain what the ASC T-ASI adds to these existing scales (see pages 2 and 3, lines 41-43 and lines 53-57).

3. On page 8 the authors should define what they mean by the terms 'soft' and 'hard drugs'

Response: In line with the reviewer’s comment, we now give definitions of soft drugs and hard drugs on page 7 lines 183-185.

4. The results section would be improved if the authors reported on the direction of the results, as well as significant differences.

Response: In line with the reviewer’s comment, we now also report the direction and statistical significance of the results in the Results section.

5. I note the authors conducted a Bonferroni correction to control error rate however given the number of comparisons why was an ANOVA not used?

Response: We conducted multiple t-tests and controlled for the increased risk of a Type I error in order to obtain insight into statistically significant differences in ASC T-ASI tariffs between respondents who did/did not use alcohol, drugs, and medicines and between respondents who were/were not delinquent. An analysis of variances (ANOVA) would have provided insight into ‘whether’ there was a difference in tariff between these groups of respondents, but not exactly ‘where’ differences occurred. By conducting multiple t-tests, we found that the tariff was statistically significantly different between respondents who did/did not use alcohol and drugs and who were/were not delinquent, but not between respondents who did/did not use medicines. 

6. I was confused by the statement that a limitation of the study were the lack of questions in the retest questionnaire, I had thought the same questionnaire was delivered at retest?

Response: We thank the reviewer for raising this point. In line with the reviewer’s comment, we clarified the second limitation of our study on page 13 (lines 278-287) in the Discussion section.

---

## [Editor Report · Decision Letter 1]

26 Aug 2020

Presentation and validation of the Abbreviated Self Completion Teen-Addiction Severity Index (ASC T-ASI): A preference-based measure for use in health-economic evaluations

PONE-D-20-03462R1

Dear Dr. Reckers-Droog,

We’re pleased to inform you that your manuscript has been judged scientifically suitable for publication and will be formally accepted for publication once it meets all outstanding technical requirements.

Kind regards,

Matthew J. Gullo

Academic Editor

PLOS ONE
---

## [Editor Report · Acceptance letter]

2 Sep 2020

PONE-D-20-03462R1 

Presentation and validation of the Abbreviated Self Completion Teen-Addiction Severity Index (ASC T-ASI): A preference-based measure for use in health-economic evaluations 

Dear Dr. Reckers-Droog:

I'm pleased to inform you that your manuscript has been deemed suitable for publication in PLOS ONE. Congratulations! Your manuscript is now with our production department. 

Kind regards, 

on behalf of

Dr. Matthew J. Gullo 

Academic Editor

PLOS ONE